# Analysis of Peri-Implant Bone Loss with a Convergent Transmucosal Morphology: Retrospective Clinical Study

**DOI:** 10.3390/ijerph19063443

**Published:** 2022-03-15

**Authors:** María Costa Castillo, Martín Laguna Martos, Rocío Marco Pitarch, Marina García Selva, Silvia del Cid Rodríguez, Carla Fons-Badal, Rubén Agustín Panadero

**Affiliations:** 1Private Practice Dentist, 46010 Valencia, Spain; macoscas@alumni.uv.es (M.C.C.); martujano@hotmail.com (M.L.M.); sildelcidr@gmail.com (S.d.C.R.); 2Department of Stomatology, Medical School of Medicine and Dentistry, University of Valencia, 46010 Valencia, Spain; rociomarco1@hotmail.com (R.M.P.); marina.garcia@uv.es (M.G.S.); rubenagustinpanadero@gmail.com (R.A.P.)

**Keywords:** infracrestal implants, crestal implants, supracrestal implants, peri-implant bone loss, convergent collar

## Abstract

Objective: The aim of this study was to analyze the peri-implant bone loss of infracrestal, supracrestal, and crestal implants from the day of placement and up to 1 year of prosthetic loading. Material and methods: A retrospective clinical study was carried out. The sample consisted of 30 implants placed on 30 patients. It was divided into three groups: infracrestal (n = 10), crestal (n = 10), and supracrestal (n = 10) implants. Results: Following the statistical analysis, it was observed that, 4 months after implant placement, the mean values of total peri-implant bone loss were 0.04 mm in infracrestal implants, 0.26 mm in crestal implants, and 0.19 mm in supracrestal implants. At the end of one year of prosthetic loading, the peri-implant bone loss was 0.12 mm in infracrestal implants, 1.04 mm in crestal implants, and 0.27 mm in supracrestal implants. It was determined that peri-implant bone loss in crestal implants was significantly higher than in supracrestal implants, and these in turn were significantly higher than in infracrestal implants. Conclusions: The implants that obtained a better biological behavior on peri-implant bone tissue were the infracrestal implants with a converging transmucosal abutment.

## 1. Introduction

Tooth loss involves both the absence of function and the impairment of the patient’s physical, social, and mental health and wellbeing [1]. A wide variety of treatments are currently available to replace missing teeth using both fixed and removable prostheses. Among them, the fixed prosthesis on implants stands out, as it is nowadays routinely performed in dental clinics, with high success rates [1].

Dental implants are devices that are anchored in the maxillary bone and act as roots which, together with the implant–prosthetic restorations, will allow us to replace a tooth, several teeth, or even all the teeth, offering the patients an improvement in their life quality, restoring their masticatory function, esthetics, and comfort, and increasing their self-esteem [2].

The consequence of tooth loss, which can complicate the prosthodontic treatment, is usually the physiological reabsorption of the bone crest, due to the cessation of the biological mechanical loads. The use of implants together with their prosthetic restorations help to restore these stresses on the maxillary bone, stimulating and preserving the bony ridges, as these will receive the mechanical forces of mastication [1,2,3].

The survival of dental implants will depend on the integration of the implant with the bone and soft tissues [4]. Peri-implant bone loss can be considered an indicator of long-term implant success. A bone loss of 0 to 0.2 mm around the implant is expected during the first year of function [5,6]. Peri-implant bone loss can be influenced by multiple factors such as occlusal stress, surgical trauma, prosthetic restoration problems, microgaps, and patient-specific risk factors [7,8,9,10].

The loss of supporting bone around an implant can compromise implant survival. As such, the management of peri-implant bone loss has become the focus of many researchers. In order to monitor follow-up controls on implants, both clinical and radiological examinations should be performed. Depending on the literature, orthopantomographic, periapical, lateral skull, and cone-beam computed tomography systems (CBCT) radiographs can be considered valid, although most experts advise the use of parallelized periapical radiographs [11].

The aim of this study was to analyze the peri-implant bone loss during the first year of functional loading on implants placed in different crown-apical positions with respect to the bone, that is, juxtacrestal, infracrestal, and supracrestal.

The working hypotheses proposed in this study were the following:

**Hypothesis** **1** **(H1).**
*The peri-implant bone loss value in crestal implants will be higher than that of supracrestal and infracrestal implants.*


**Hypothesis** **2** **(H2).**
*There will be no significant differences in peri-implant bone loss between supracrestal and infracrestal implants.*


## 2. Materials and Methods

### 2.1. Study Design

A retrospective clinical study was carried out with a 1-year follow-up, with the aim of evaluating peri-implant bone loss after functional loading of implants, in both maxillary and mandibular premolar areas. The implants were placed by the same operator at the Dental Clinic of the Faculty of Medicine and Dentistry of the University of Valencia, between 2018 and 2019. The data were collected after signing the informed consent, approved by the Human Research Ethics Committee of the University of Valencia (registration no.: 1500285).

The sample used consisted of 30 implants placed on 30 patients. The sample was divided into three groups: infracrestal implants (n = 10), crestal implants (n = 10), and supracrestal implants (n = 10).

The implants used were the following (Table 1, Table 2 and Table 3):

### 2.2. Morphological Characteristics of the Implants

Three different types of implants were used in this study, all of them from the company Sweden & Martina (Padua, Italy). They are described in detail below.

Supracrestal implants (PRAMA) present a cylindrical morphology. The transmucosal collar is characterized by a 0.80 mm high cylindrical portion followed by a 2.00 mm high hyperbolic geometric portion, designed to guarantee a reliable continuity with the abutment (Figure 1).

Infracrestal implants (Shelta) present a tapered morphology which decreases progressively as the implant length increases. The angle remains stable between implants of different diameters but of equal length. All Shelta implants are characterized by a collar with a microsurface treatment with micro-stripes (UTM) of 1.00 mm in height (Figure 2).

These implants require the use of abutments, known as XA abutments, manufactured in titanium with a conical shape and presence of micro-grooves at the base, with a morphology lacking a termination line. They present a single diameter of 3.80 mm, with heights of 5.50 mm and 6.50 mm (Figure 3).

XA abutments offer the possibility of working with the “one abutment one time” protocol, so that the abutment is placed on the day of surgery, thus avoiding the connections and disconnections that are so harmful to the stability of the soft tissue.

Crestal implants (Premium) present a machined collar of 0.80 mm of height, a cylindrical body, and a thread of the spiral with a pitch of 1.00 mm and 0.40 mm of depth (Figure 4), except for the implants with a diameter of 3.30 mm, which have a thread pitch of 0.60 mm and a depth of 0.30 mm (Figure 5).

### 2.3. Inclusion and Exclusion Criteria

The inclusion criteria were the following: Patients over 18 years old, that had signed the informed consent, presenting supracrestal, infracrestal, and crestal implants in the premolar area. In order to be included in the present study, the following records had to be available: parallelized periapical radiographs from the day of implant placement surgery, the day of crown placement, and after 12 months of prosthetic loading. Furthermore, patients had to be of overall good health and non-smokers.

The exclusion criteria applied were the following: Patients under 18 years old, those patients who did not sign the informed consent form, implants with a functional load under one year, those for whom follow-up periapical radiographs were not available, or those with guided bone regeneration in the area.

Three different types of implants were chosen, Prama (supracrestal), Shelta (infracrestal), and Premium (crestal), from Sweden & Martina (Padua, Italy). All the implants had an implant-supported, screw-retained metal–ceramic crown. In the infracrestal group, a transmucosal abutment of conical morphology was placed on the day of implant surgery to prevent the prosthetic crown–implant interface from being infraosseous (XA abutment; Sweden & Martina).

### 2.4. Clinical Procedure

The inclusion criteria were the following: Patients over 18 years old, that had signed the informed consent, presenting supracrestal, infracrestal, and crestal implants in the premolar area. In order to be included in the present study, the following records had to be available: parallelized periapical radiographs from the day of implant placement surgery, the day of crown placement, and after 12 months of prosthetic loading. Furthermore, patients had to be of overall good health and non-smokers.

The exclusion criteria applied were the following: Patients under 18 years old, those patients who did not sign the informed consent form, implants with a functional load under one year, those for whom follow-up periapical radiographs were not available, or those with guided bone regeneration in the area.

### 2.5. Measurement Process

Marginal bone loss was assessed on intraoral, parallelized periapical radiographs. Initial radiographs were collected on the implant placement day, on the prosthetic restoration placement day, and on the revision day after one year of loading. The radiographs were taken using phosphor radiographic plates, with the aid of a Rinn System XPC intraoral radiograph positioner (Denstsply, St. Charles, IL, USA). The DBSWIN software was used to collect the radiographs. The analysis of the radiographs was performed by a single observer, with the purpose of reducing the probability of carrying out different assessments and thus reducing the bias between measurements.

Rhinoceros (Robert Mcneel & Associates, Seattle, WA, USA), a CAD software for computer-aided design and 3D surface modeling, was used to perform the analysis of the radiographs. The radiographs were scaled using as a value the width (in millimeters) of the prosthetic platform of the implant. In this manner, distortion errors were avoided.

The bone level was measured as the linear distance between the implant–abutment interface and the level of the crestal bone margin, mesial and distal to each implant, all calculated in millimeters. Therefore, this distance was analyzed both mesially and distally on the day of implant placement surgery. Subsequently, the distance of the same points was measured on the radiograph on the day of prosthetic restoration placement. At this point, the first bone loss figures were obtained by calculating the difference between the heights recorded. The same measurements were taken, but on the control radiographs, after one year of prosthetic loading, and the peri-implant bone loss was calculated in relation to the day of implant placement surgery. At this point, the second bone loss figures for the study were obtained (Figure 6A–C).

## 3. Results

The main variable was marginal bone loss, measured in the mesial and distal zones of the implant.

The descriptive analysis contains the most relevant statistical values for marginal bone loss (MBL): mean, standard deviation, minimum, maximum, and quartiles. The inferential analysis aims to determine whether there are differences in peri-implant bone loss according to each group, as well as to compare the results at mesial and distal levels.

In each measurement area (mesial, distal, and total), a nonparametric Brunner–Langer model was estimated for longitudinal data within subjects in the group and, between subjects, the measurement time. The value of the ATS statistic was used to conclude on main effects and interactions.

For multiple comparisons between groups at a specific time, the Mann–Whitney test with Bonferroni correction was used. For the comparison of peri-implant bone loss between zones, a new Brunner–Langer model was estimated with this new within-subject factor.

The significance level used in the analyses was 5% (α = 0.05).

The proposed statistical methodology, with a confidence level of 95% and considering an effect size to detect f = 0.4 (large), reached a power of 57.7% for the differences between groups. In order to detect changes in peri-implant bone loss over time, the power was increased to 99.1%.

The sample used consisted of 30 implants placed on 30 patients. The sample was divided into three groups: infracrestal implants (n = 10), crestal implants (n = 10), and supracrestal implants (n = 10). After radiographic analysis, three tables (Table 4, Table 5 and Table 6) were elaborated, one for each implant group, detailing the amount of peri-implant marginal bone at two specific times:−T1: The day of prosthesis placement (4 months after surgery).−T2: After 1 year of prosthetic loading.

The intention at these specific intervals was to examine the overall peri-implant bone loss in each of the groups, analyzing it statistically, comparing the first time, 4 months after implant placement, specifically, the day of prosthetic restoration placement, and the second time, 1 year after prosthetic loading, and making a comparison between the groups that constitute the sample.

In the first interval (4 months after implant placement), the loss mean and median was 0.04 mm in infracrestal implants (interquartile range (IQR): 0.03 ± 0.11), 0.26 mm (IQR: 0.00 ± 0.43) in crestal implants, and, finally, 0.19 mm in supracrestal implants (IQR: 0.18 ± 0.23). With the results obtained through the Brunner–Langer model test, a significant peri-implant bone loss was recorded (*p* < 0.001); being statistically significant in infracrestal (*p* < 0.001), crestal (*p* = 0.001), and supracrestal (*p* < 0.001) implants, although the progression was similar in the three groups (*p* = 0.142).

At the second interval (after 1 year of prosthetic loading), the mean and median loss was 0.12 mm in infracrestal implants (IQR: 0.09 ± 0.16), 1.04 mm in crestal implants (IQR: 0.90 ± 1.09), and, finally, 0.27 mm in supracrestal implants (IQR: 0.22 ± 0.35). The peri-implant bone loss was significant in any of the three groups (*p* < 0.001). Regarding the differences between groups, marginal bone loss was significantly higher in the crestal implants than that observed in the supracrestal implants (*p* < 0.001), and these in turn were higher than that regarding the infracrestal implants (*p* < 0.001). These data were extracted from the results of the Mann–Whitney test with Bonferroni correction.

With regard to the analysis of bone loss at the mesial level, at the first interval, the mean and median loss was 0.06 mm in infracrestal implants (IQR: 0.04 ± 0.07), 0.24 mm in crestal implants (IQR: 0.00 ± 0.054), and, finally, 0.19 mm in supracrestal implants (IQR: 0.12 ± 0.25). Regarding the differences between groups, when observing the results of the Mann–Whitney test with Bonferroni correction, the only differences were reached in the comparison between infracrestal and supracrestal implants (*p* < 0.001). The infracrestal implants presented lower levels of peri-implant bone loss with respect to the supracrestal implants (Figure 7).

At the second time interval, the mean and median bone loss was 0.12 mm (IQR: 0.07 ± 0.13) in infracrestal implants, 1.01 mm (IQR: 0.88 ± 1.42) in crestal implants, and, finally, 0.25 mm in supracrestal implants (IQR: 0.19 ± 0.37). Peri-implant bone loss was statistically significant in any of the three groups (*p* < 0.001). Regarding the differences between the implants, the results of the Mann–Whitney test with Bonferroni correction determined that the peri-implant bone loss in the crestal implants was significantly higher than that of the supracrestal implants (*p* < 0.001) and, in turn, the latter lost more bone than the infracrestal implants (*p* = 0.004). The infracrestal implants lost less bone than the crestal implants, and this difference was statistically significant (*p* < 0.001) (Figure 8).

In addition, the analysis of peri-implant bone loss at the distal level was also studied, and the results obtained were as follows:

At the first time interval, the means and medians of bone loss were 0.03 mm (IQR: 0.02 ± 0.11) in infracrestal implants, 0.22 mm (IQR: 0.00 ± 0.50) in crestal implants, and 0.17 mm (IQR: 0.11 ± 0.23) in supracrestal implants. Regarding the differences between groups, through the results of the Mann–Whitney test with Bonferroni correction, differences were only reached in the comparison between infracrestal and supracrestal implants (*p* = 0.006). Infracrestal implants expressed less peri-implant bone loss with respect to supracrestal implants (Figure 9).

At the second time interval, the means and medians regarding bone loss were 0.13 mm (IQR: 0.03 ± 0.15) for infracrestal implants, 0.84 mm (IQR: 0.67 ± 1.25) for crestal implants, and, finally, 0.25 mm (IQR: 0.22 ± 0.33) for supracrestal implants. All figures are quite similar to those obtained for the mesial zone. The peri-implant bone loss was statistically significant in any of the three groups (*p* < 0.001). Regarding the differences between classes, by means of the Mann–Whitney test with Bonferroni correction, it was determined that peri-implant bone loss in crestal implants was significantly higher than that of the supracrestal implants (*p* = 0.006) and these, in turn, were higher in comparison to the infracrestal implants (*p* = 0.003). Peri-implant bone loss in the infracrestal implants was lower than in the supracrestal implants (*p* = 0.009) (Figure 10).

The results above demonstrated a very similar pattern in the progression of peri-implant bone loss in both the mesial and distal zones. All statistical tests performed offered the same result in both zones. An even more general model was estimated by including the factor “measurement zone” in order to be able to formally demonstrate that this aspect does not induce relevant heterogeneity. By virtue of the results of the ATS test of the Brunner–Langer model, no significant factor involving the zone was determined, because the zone, as can be seen, has no influence on the evolution of bone loss. In the graph below, it can be observed that the results for the mesial and distal zones practically overlap (Figure 11).

## 4. Discussion

Radiographic evaluation is one of the most widely used diagnostic methods in implant dentistry, as it allows bone availability to be monitored. Specifically in this study, periapical radiographs were used, with the help of a standardized parallelization technique that provides greater clarity of the area studied, and, in consequence, the progressive loss of bone tissue around the implant can be determined. As in this study, there exist several other studies in literature that have also used this same method, such as Pellicer-Chover [12], who used the method of the Rinn system positions (Dentsply) as well. Apart from using parallelization methods, the Rhinoceros program was also emplaced, which is a software that helps prevent the inherent distortion of intraoral radiographs. In the present study, the reference used was the diameter (in millimeters) of the prosthetic platform of the implant, which is provided by the implant manufacturer, in the same way as Marconcini et al. [13] and Canullo et al. [14] did. On the contrary, studies such as Pellicer-Chover et al. [12] took the length of the implant as a reference.

Certain studies use panoramic radiographs for analyzing peri-implant bone loss, taking into account that these have an inferior definition and that they distort the images and overlap bony structures of the spine. Moreover, the magnification in each region is inconsistent and the reduced resolution produces a greater risk of measurement loss [15], as was the case of Galindo et al. [16], who obtained standardized digital panoramic radiographs. It is also possible to find the use of cone-beam computed tomography in several investigations to quantify peri-implant bone loss, as, for example, in the case of Hadzik et al. [17].

The results obtained in the present study, taking into account that peri-implant bone loss was analyzed in two different time intervals, were the following: In the first time interval, 4 months after surgery, the mean values were 0.04 mm in infracrestal implants, 0.26 mm in crestal implants, and, finally, 0.19 mm in supracrestal implants; while in the second time interval, after one year of prosthetic loading, the results obtained were 0.12 mm in infracrestal implants, 1.04 mm in crestal implants, and, finally, 0.27 mm in supracrestal implants. In 2019, Pico et al. [18] performed a study on 66 implants, divided into two groups: one group with abutments of 3 mm in height and implants placed 2 mm subcrestally, and another group with abutments of 1 mm in height and implants placed crestally, in which they obtained values similar to the ones in this study since, after one year of follow-up, the subcrestal implants had a mean peri-implant bone loss of 0.12 mm and the crestal implants had a mean peri-implant bone loss of 0.95 mm. Other authors, such as Marconcini et al. [13], studied 36 implants (implants with tapered abutments) and obtained similar results to this study, since their implants showed a mean peri-implant bone loss of 0.18 mm after 1 year of prosthetic loading. In contrast, other authors obtained lower values than those in the present study, such as, for example, Canullo et al. [19], who studied 16 implants at the tissue level with a convergent collar, and their mean peri-implant bone loss was 0.071 mm at 3 years of follow-up. Therefore, these values are more favorable than those of the present study. In the case of Pellicer-Chover et al. [12], they studied a total of 265 implants (implants at the level of the osseous crest and implants 2 mm below the crest). The implants at the level of the crest presented a mean peri-implant bone loss of 0.29 mm, and in the subcrestal implants the value was 0.09 mm at 36 years of follow-up; values lower than ours.

In the present study, significant differences regarding the peri-implant bone loss were observed in the three groups of implants analyzed. In the crestal implants, the peri-implant bone loss was significantly higher than in the supracrestal implants and these, in turn, were higher than in the infracrestal implants. Therefore, the first hypothesis (H0) is accepted; which stated that the value of peri-implant bone loss in crestal implants would be greater in comparison with supracrestal and infracrestal implants. However, the second hypothesis (H1) is rejected since it stated that there would be no significant differences in peri-implant bone loss between supracrestal and infracrestal implants, but nevertheless statistically significant differences were found. Regarding the measurement zone, the results of all the statistical tests performed proved a very similar pattern in the progression of peri-implant bone loss both mesially and distally; therefore, the fourth hypothesis (H3), which stated that there would be no significant differences in peri-implant bone loss of the implant, both mesially and distally, is accepted.

A study which obtained similar results was that of Canullo et al. [14] in 2021, who found a statistically significant difference between the tissue-level group and the bone-level group with respect to bone reabsorption, with the tissue-level group obtaining the better result. Likewise, Pico et al. [18] analyzed two groups of implants, one group with abutments 3 mm high and implants placed 2 mm subcrestally, and another group with abutments 1 mm high and implants placed crestally, with a follow-up up to 12 months after surgery. The statistical analysis revealed significant differences between the groups. A greater bone preservation was observed in the group of implants placed subcrestally than in the group of crestal implants.

Similarly, in the study by Agustín et al. (2019) [20], 120 implants were analyzed divided into six groups, three of them placed at the level of the osseous crest (internal hexagonal connection, hexagonal and anti-rotational connection, and internal hexagonal connection with peripheral sealing) and the other three groups placed at supracrestal level (convergent implant with machined collar and internal hexagon, divergent polished collar and internal hexagon connection, and implant with divergent polished collar and internal octagonal connection). The results showed a statistically significant difference, with greater peri-implant bone loss around implants placed at crest level. They concluded the study by stating that peri-implant bone loss is influenced by the level of implant placement in relation to the bone crest and by the morphology of the prosthetic platform.

In the present study, significant differences were obtained between the infracrestal and supracrestal implants in comparison with the crestal implants. In consequence, it can be stated that part of this difference is influenced by the design of the transmucosal neck, being that the supracrestal implants present a transmucosal neck that is characterized by having a cylindrical portion of 0.80 mm in height, followed by a geometric hyperbolic portion of 2.00 mm in height, and the infracrestal implants presented a neck with a microsurface treatment with micro-rays of 1.00 mm in height and with an XA abutment, unlike the crestal implants that presented a machined neck of 0.80 mm in height. For this reason, the third hypothesis (H2) is accepted, which proposed that the transmucosal neck design, with convergent shape, of the tissue level implants would improve the stability of the bone tissue in comparison to the crestal implants.

In the case of Spinato et al. [21], with a total of 51 implants (25 implants without platform changes (control group) and 26 implants with a switched platform (test group)), the results showed that the mean mesial and distal peri-implant bone loss at 12 months was always greater in the control group than in the test group, with a statistical significance. As a conclusion, they highlighted the significant inverse relationship between marginal bone loss and the height of the corresponding abutment. In a similar study by Montemezzi et al. [22], the sample was composed of 122 dental implants divided into two groups (wide-neck rough implants and reduced-neck rough implants) where the analysis revealed a relationship between neck design and peri-implant bone loss, so that marginal bone loss was generally lower for implants with wide necks than for those with reduced necks. It is important to note that neck design was statistically significant. Similarly, Agustín et al. (2021) [23] placed a total of 120 implants divided into two groups. The first group was composed of implants with convergent transepithelial tissue-level collars, and the second group were bone-level implants, among which significant differences were found regarding peri-implant bone loss. Tissue-level implants presented less bone loss compared to bone-level implants, regardless of the type of prosthesis.

However, there are studies such as that of Pellicer-Chover et al. [12], who had a sample of 265 implants (implants at the level of the osseous crest and implants 2 mm below the crest), but their results did not show statistically significant associations between peri-implant clinical parameters and the apico-coronal placement position of the implant with respect to the osseous crest. On the contrary, they observed significantly less bone loss in subcrestal implants compared to crestal implants. They concluded that minimal peri-implant bone loss was reflected regardless of the implant placement technique used. With similar results to this study, a randomized trial by Lago et al. [24] was identified in which 50 tissue-level implants and 50 bone-level implants were compared. In this trial, both types of implants showed good results with respect to crestal bone level maintenance with no significant differences between the two types of implants after 3 years of follow-up.

The review by Bennardo F. et al. [25] studied the use of magnetic mallets in oral surgery and implant procedures in terms of tissue healing, surgical outcome, and complication rate in comparison with traditional instruments, which were used in the present study. The results obtained proved that the survival rate of the implants was 98.9% in the groups performed with magnetic mallet and 95.42% in the control groups, which were performed with traditional instruments.

A small sample size and a limited follow-up time were certain limitations of the present study.

## 5. Conclusions

Despite the limitations of the present study, it may be concluded that:−The implants with the best results with regard to peri-implant bone loss were the infracrestal implants, followed by the supracrestal implants, and, lastly, by the crestal implants.−The peri-implant bone loss values obtained after the first year of prosthetic loading are within the range considered acceptable in the literature reviewed.−The transcrestal morphology of the implants has a significant influence on peri-implant bone loss, since infracrestal and supracrestal implants with convergent transmucosal necks present lower peri-implant bone loss values compared to crestal implants with cylindrical transmucosal necks with parallel walls.

Further studies are required to confirm the authors’ hypothesis.

## Figures and Tables

**Figure 1 ijerph-19-03443-f001:**
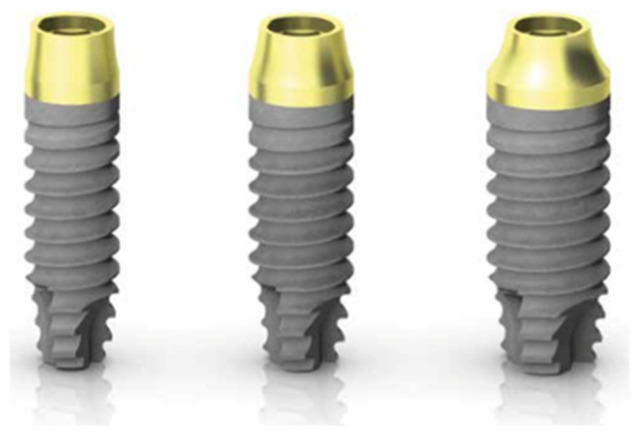
Supracrestal implant (PRAMA, Sweden & Martina).

**Figure 2 ijerph-19-03443-f002:**
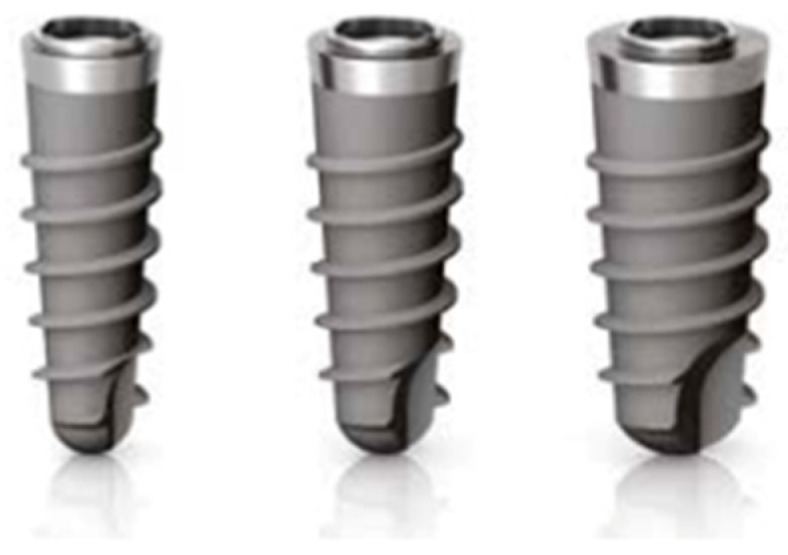
Implante infracrestal (SHELTA, Sweden & Martina).

**Figure 3 ijerph-19-03443-f003:**
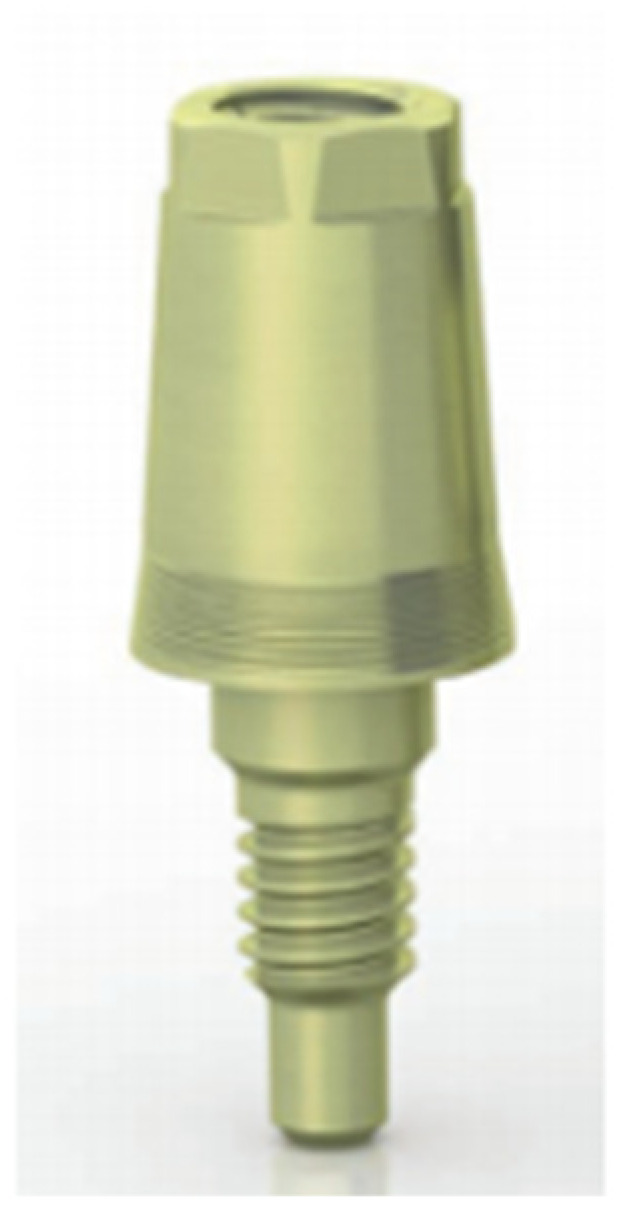
XA abutments (Sweden & Martina).

**Figure 4 ijerph-19-03443-f004:**
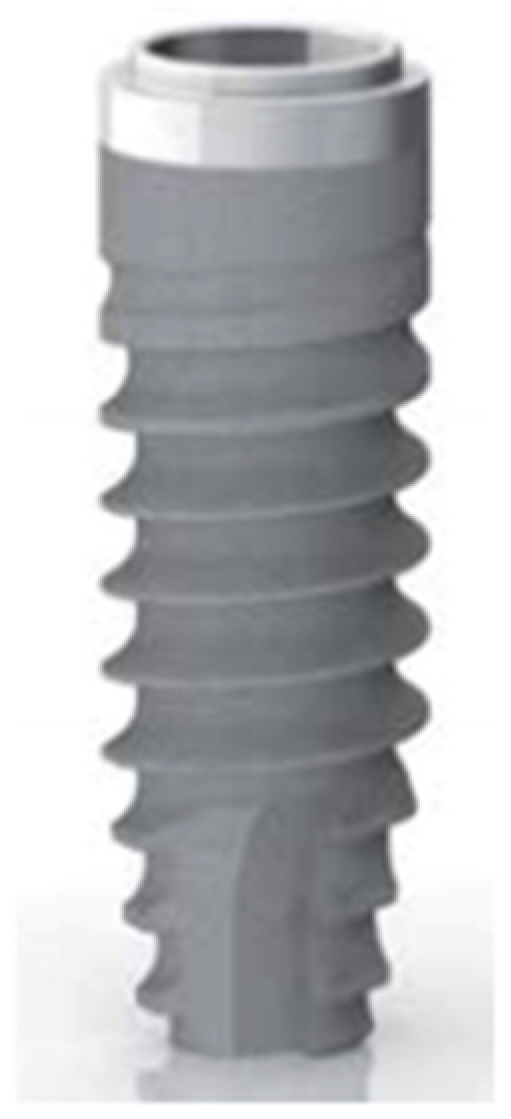
Crestal implant (PREMIUM, Sweden & Martina).

**Figure 5 ijerph-19-03443-f005:**
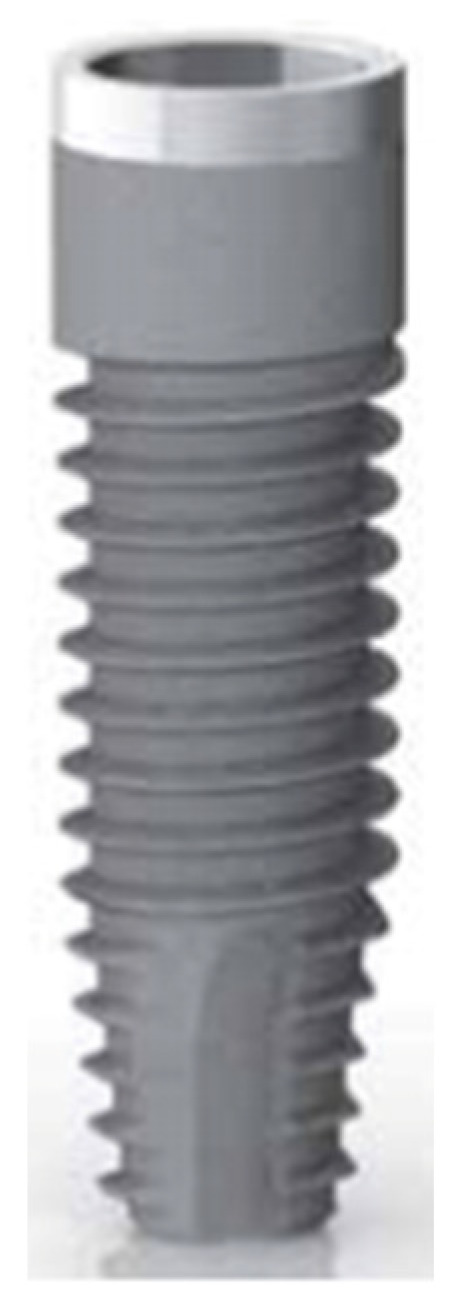
Crestal implant (PREMIUM 3.30 mm, Sweden & Martina).

**Figure 6 ijerph-19-03443-f006:**
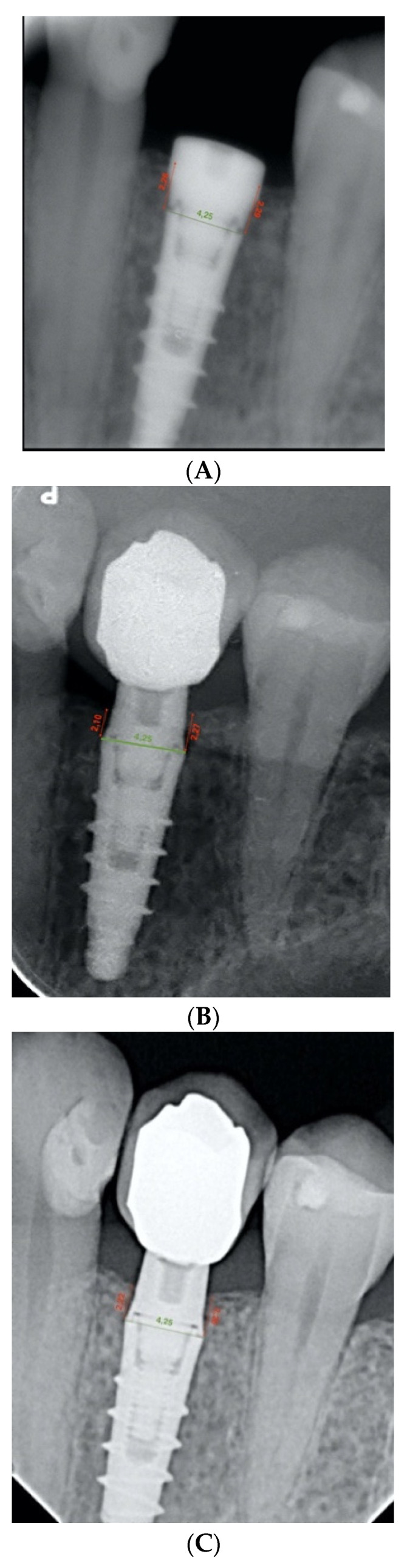
(**A**) Periapical radiograph of an infracrestal implant (Shelta) on the day of surgery. (**B**) Periapical radiograph of an infracrestal implant (Shelta) after 1 year of prosthetic loading. (**C**) Periapical radiograph of an infracrestal implant (Shelta) on the day of prosthetic placement, 4 months after surgery.

**Figure 7 ijerph-19-03443-f007:**
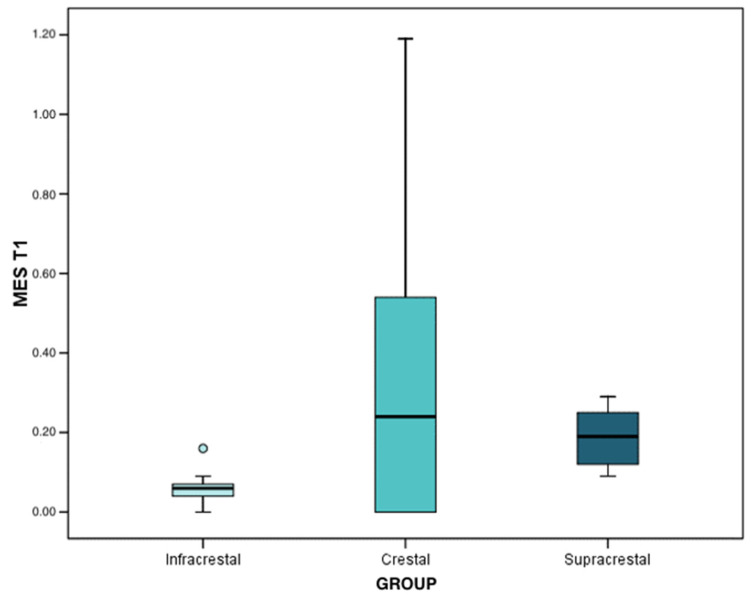
Mesial bone loss at T1 (4 months after implant placement) according to each group.

**Figure 8 ijerph-19-03443-f008:**
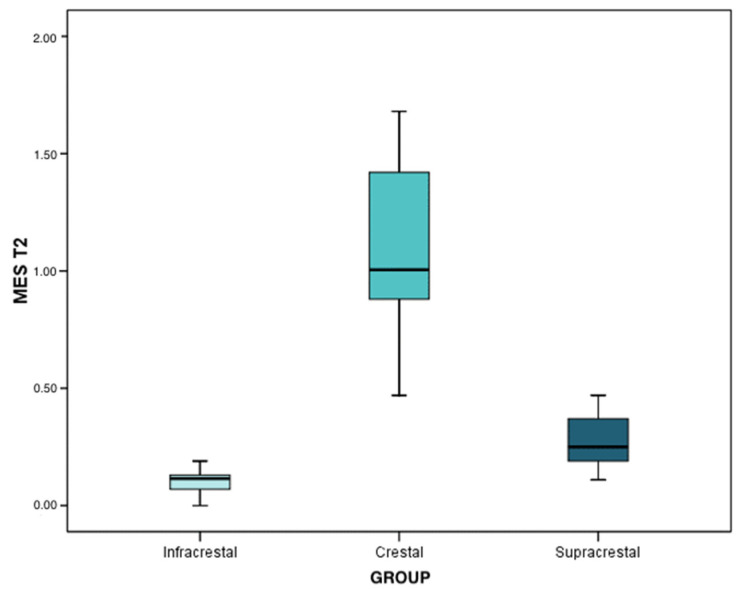
Mesial bone loss in T2 (after 1 year of prosthetic loading), according to each group.

**Figure 9 ijerph-19-03443-f009:**
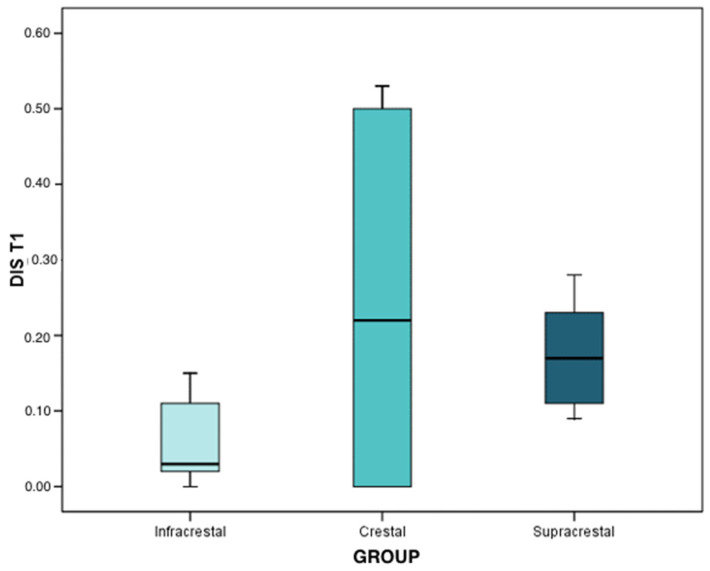
Distal bone loss at T1 (4 months after implant placement), according to each group.

**Figure 10 ijerph-19-03443-f010:**
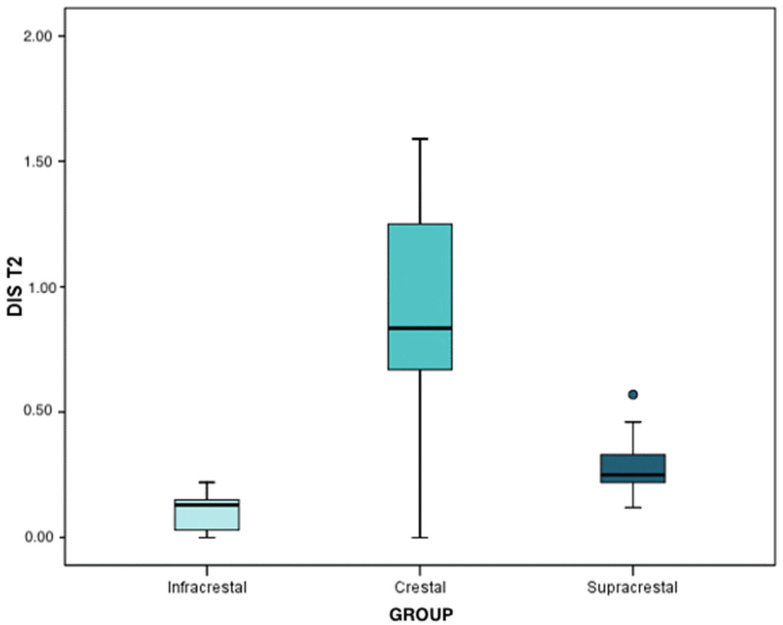
Distal bone loss at T2 (after 1 year of prosthetic loading), per group.

**Figure 11 ijerph-19-03443-f011:**
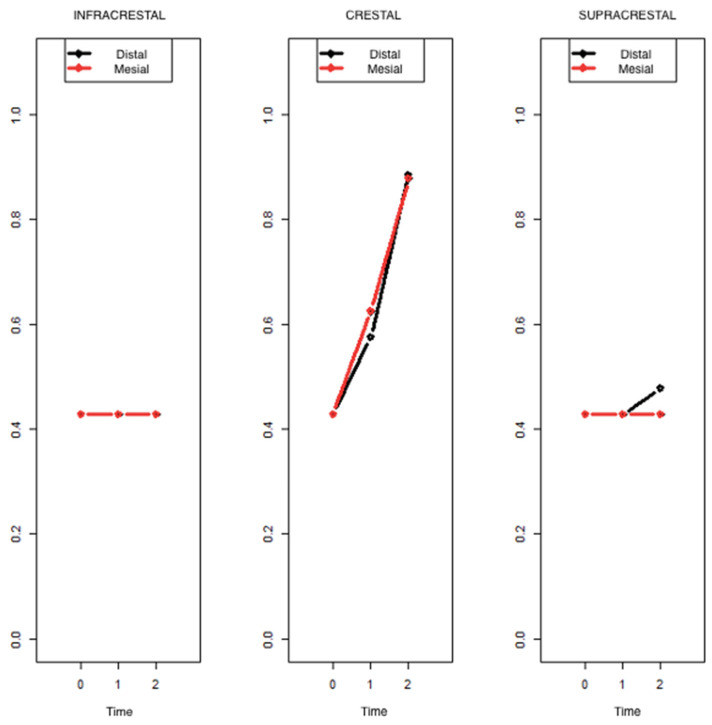
Results of the changes in the bone crest for mesial and distal zones.

**Table 1 ijerph-19-03443-t001:** Diameter and length in millimeters of infracrestal implants (SHELTA, Sweden & Martina).

Infracrestal (Number of Implants)	Diameter (mm)	Length (mm)
2	3.8	10
2	3.8	11.5
5	4.25	11.5
1	4.25	13

**Table 2 ijerph-19-03443-t002:** Diameter and length of crestal implants (PREMIUM, Sweden & Martina).

Crestal (Number of Implants)	Diameter (mm)	Length (mm)
1	3.3	11.5
2	3.80	11.5
2	4.25	8.5
3	4.25	10
2	4.25	11.5

**Table 3 ijerph-19-03443-t003:** Diameter and length of supracrestal implants (PRAMA, Sweden & Martina).

Supracrestal (Number of Implants)	Diameter (mm)	Length (mm)
1	3.8	11.5
2	4.25	8.5
2	4.25	10
4	4.25	11.5
1	5	10

**Table 4 ijerph-19-03443-t004:** Peri-implant bone loss at different times calculated in infracrestal implants (Shelta; Sweden & Martina).

Infracrestal Implants	Mesial Area of the Implant		Distal Area of the Implant	
	Bone Loss 3 Months after Implant Placement (Day of Prosthesis Placement) (mm)	Loss after 1 Year of Prosthesis Placement (mm)	Bone Loss 3 Months after Implant Placement (Day of Prosthesis Placement) (mm)	Loss after 1 Year of Prosthesis Placement (mm)
1	−0.04	−0.16	−0.02	−0.02
2	−0.16	−0.19	−0.11	−0.14
3	−0.07	−0.07	−0.03	−0.12
4	−0.05	−0.12	−0.02	−0.15
5	−0.06	−0.13	−0.11	−0.15
6	−0.06	−0.12	−0.15	−0.19
7	−0.06	−0.10	−0.01	−0.12
8	0.00	0.00	−0.03	−0.03
9	0.00	−0.01	0.00	0.00
10	−0.09	−0.11	−0.12	−0.22

**Table 5 ijerph-19-03443-t005:** Peri-implant bone loss at different times calculated in crestal implants (Premium; Sweden & Martina).

Crestal Implants	Mesial Area of the Implant		Distal Area of the Implant	
	Bone Loss 3 Months after Implant Placement (Day of Prosthesis Placement) (mm)	Loss after 1 Year of Prosthesis Placement (mm)	Bone Loss 3 Months after Implant Placement (Day of Prosthesis Placement) (mm)	Loss after 1 Year of Prosthesis Placement (mm)
1	0.00	−0.88	−0.50	−0.77
2	−0.48	−1.42	0.00	−0.67
3	0.00	−0.62	−0.53	−1.55
4	−0.54	−1.00	−0.28	−1.11
5	0.00	−0.47	0.00	−1.59
6	−1.19	−1.68	−0.16	−0.51
7	−0.98	−1.34	−0.52	−0.71
8	0.00	−0.9	0.00	−0.9
9	0.00	−1.01	0.00	0.00
10	−0.52	−1.46	−0.34	−1.25

**Table 6 ijerph-19-03443-t006:** Peri-implant bone loss at different times calculated in supracrestal implants (Prama; Sweden & Martina).

Supracrestal Implants	Mesial		Distal	
	Bone Loss 3 Months after Implant Placement (Day of Prosthesis Placement) (mm)	Loss after 1 Year of Prosthesis Placement (mm)	Bone Loss 3 Months after Implant Placement (Day of Prosthesis Placement) (mm)	Loss after 1 Year of Prosthesis Placement (mm)
1	−0.25	−0.29	−0.26	−0.28
2	−0.23	−0.37	−0.23	−0.33
3	−0.11	−0.19	−0.28	−0.28
4	−0.19	−0.28	−0.17	−0.22
5	−0.12	−0.12	−0.09	−0.12
6	−0.09	−0.11	−0.09	−0.22
7	−0.28	−0.44	−0.11	−0.13
8	−0.19	−0.22	−0.17	−0.22
9	−0.19	−0.22	−0.17	−0.57
10	−0.29	−0.47	−0.19	−0.46

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
