# Peer review of "Analysis of Peri-Implant Bone Loss with a Convergent Transmucosal Morphology: Retrospective Clinical Study"

_ijerph, 2022, doi:10.3390/ijerph19063443_

Round 1

Reviewer 1 Report

Thank you for giving me the opportunity to review the manuscript.
After reviewing all the text, this reviewer recommends a thorough review of the content of the manuscript since the English language must be completely reviewed within the text and the technical and cientific vocabulary!

You can't just translate from one language to another! this is unacceptable in a scientific writing
In the current state, the authors must consider whether the information described is precise enough to be a reproducible study.
The reviewer thinks that with the way it is written it is a true retrospective study or if it is a randomly collected group of patients.

Title:

Please fully rewrite the title, Perimplant bone loss instead of bone loss and subrecestal placed implants

Abstract

What is hard tissue?

The statistical analysis is not necessary in the abstract section

Introduction

Too much information regarding peri-implant bone loss but without study rationale or justification.

The null hypothesis should be rewritten.

The objecive is not well described!

Material and methods

This section is really worrying since the information provided is very very limited, without explaining the devices used or the surgical technique in an appropriate manner, much less the prosthodontic phase. There is a lack of English technical language in this section!!!!

In a surgical phase? what does that mean?

Legal age? when? it depends on the country!

No info about the placement and loading protocol!!!!!

Surgical and Prosthodontic information is lacking!!!!

Please include and describe the clinical outcomes!

How do you ensure the periodical radiographs were in the same position for the further analysis? 

THE DESCRIPTIVE ANALYSIS SHOULD BE INCLUDED IN THE RESULTS SECTION AND NOT IN THE M&M.

Please include if the hypothesses wehre recjected or accepted in the discussion section!

Plese include the straight and study weknessess or limitations.

On the basis? Based on?

Author Response

First of all thank you for your suggestions to improve our article, below I attach the changes we have made based on your recommendations

1.Title: Please fully rewrite the title, Perimplant bone loss instead of bone loss and subrecestal placed implants

We have made changes in the title of our article, here below I show them to you:

“Analysis of perimplant bone loss with a Convergent Transmucosal Morphology: Retrospective clinical study”

  1. Abstract: What is hard tissue? The statistical analysis is not necessary in the abstract section

Thank you for pointing this out. We have made the change from "hard tissue" to "bone tissue"

  1. Introduction: Too much information regarding peri-implant bone loss but without study rationale or justification. The null hypothesis should be rewritten. The objecive is not well described!

Thanks for these notes, we have made the pertinent changes, for example, I attach here the rewritten objective

Abstract: The aim of this study was to analyze the bone loss of infracrestal, supracrestal and crestal implants from the day of placement and up to 1 year of prosthetic loading…”

  1. Material and methods

This section is really worrying since the information provided is very very limited, without explaining the devices used or the surgical technique in an appropriate manner, much less the prosthodontic phase. There is a lack of English technical language in this section!!!! In a surgical phase? what does that mean?

Thank you for your note, we have made the necessary changes in the language.

  1. Legal age? when? it depends on the country!

Thank you for pointing this out. We have changed legal age for older than 18 years-old.

  1. No info about the placement and loading protocol!!!!! Surgical and Prosthodontic information is lacking!!!! Please include and describe the clinical outcomes!

Thank you for your suggestions, we have added the information in the corresponding section, here I attach part of the modification.

“2.4. Clinical procedure

All implants were placed by the same operator between 2018 and 2019. They were placed in posterior areas, specifically in the premolar area, both in the maxilla and mandible, following the same surgical protocol. All surgeries were performed under local anesthesia (4% articaine with adrenaline 1:100,000 (Inibsa®, Lliça de Vall, Barcelona, Spain). The implants were placed in the posterior area, specifically in the premolar area, preparing the implant bed using the drill and the drilling protocol recommended by the implant manufacturer. The surgical incision in the mucosa was supracrestal. All patients were treated in a single surgery session by placing healing abutments on the prosthetic platforms of the implants. After implant placement and suturing, each patient received 500 mg amoxicillin (Clamoxyl®, GlaxoSmithKline, Madrid, Spain) three times a day for 7 days, 600 mg ibuprofen (Bexistar®, Laboratorio Bacino, Barcelona, Spain) to take as needed, and 0.12 % chlorhexidine mouthwash (GUM®, John O. Butler/Sunstar, Chicago, IL, USA) twice a day for two weeks. Treatment with chlorhexidine-containing dentifrice was also recommended. Sutures were removed 8-10 days after surgery.

Following surgery, all the patients attended a check-up after one week, one month and, finally, after four months, to take measurements of the definitive crown.

Four months after implant placement surgery, the patients were scheduled for a digital measurement of the final prosthesis using a Trios intraoral scanner (3 Shape, Copenhagen, Denmark).

All crowns placed were screw-retained implant-supported crowns, made of chrome-cobalt metal (Cr-Co) and feldspathic ceramic coating; they were fabricated using computer-aided design and computer-aided manufacturing (CAD-CAM) software.”

  1. How do you ensure the periodical radiographs were in the same position for the further analysis? 

Specifically, in this study, we use periapical radiographs, with the help of a standardized parallelization technique that will provide us with greater clarity of the area we want to study, and with them we can determine the progressive loss of bone tissue around the implant. In this way it also helps us to ensure that they are as similar as possible to each other.

  1. THE DESCRIPTIVE ANALYSIS SHOULD BE INCLUDED IN THE RESULTS SECTION AND NOT IN THE M&M.

Thanks for your suggestions, we have added the statistical analysis in the results and not in material and methods as it was previously.

  1. Please include if the hypothesses wehre recjected or accepted in the discussion section!

Thanks for the note, we have added it, I attach the modification.

“In the present study significant differences regarding the peri-implant bone loss were observed in the three groups of implants analyzed; in the crestal implants the loss was significantly higher than in the supracrestal implants and these, in turn, were higher than in the infracrestal implants. Therefore, the first hypothesis (H0) is accepted; which stated that the value of peri-implant bone loss in crestal implants would be greater in comparison with supracrestal and infracrestal implants. However, the second hypothesis (H1) is rejected since it stated that there would be no significant differences in peri-implant bone loss between supracrestal and infracrestal implants, but nevertheless statistically significant differences were found. Regarding the measurement zone, the results of all the statistical tests performed proved a very similar pattern in the progression of bone loss both mesially and distally; therefore, the fourth hypothesis (H3), which stated that there would be no significant differences in peri-implant bone loss mesially or distally of the implant, is accepted.”

“In our study, significant differences were obtained between the infracrestal and supracrestal implants in comparison with the crestal implants. In consequence, it can be stated that part of this difference is influenced by the design of the transmucosal neck, being that the supracrestal implants present a transmucosal neck that is characterized by having a cylindrical portion of 0.80 mm in height, followed by a geometric hyperbolic portion of 2.00 mm in height and the infracrestal implants used presented a neck with a microsurface treatment with micro-rays of 1.00 mm in height and with an XA abutment, unlike the crestal implants that presented a machined neck of 0.80 mm in height. For this reason, we accept the third hypothesis (H2), which proposed that the transmucosal neck design, with convergent shape, of the tissue level implants, would improve the stability of the bone tissue in comparison to the crestal implants.“

  1. Plese include the straight and study weknessess or limitations.

Thanks for the note, we have added it, I attach the modification.

“A small sample size and a limited follow-up time were certain limitations of the present study.”

  1. On the basis? Based on?

Thank you for the note, we have made a change in the phrase.

“Despite the limitations of the present study, it may be concluded that: “

Reviewer 2 Report

Manuscript ID: ijerph-1603116

Title: Analysis of bone loss in supracrestal implants with a convergent transmucosal morphology

1.What is the main question addressed by the research?

To evaluate peri-implant bone loss after one year of functional loading of supracrestal implants with a convergent transmucosal morphology.

2.Is it relevant and interesting?

The article is relevant and interesting.

3.How original is the topic?

The topic is current.

4.What does it add to the subject area compared with other published material?

The authors have collected and analyzed a great deal of data.

5.Is the paper well written?

Yes, the article is well written.

6.Is the text clear and easy to read?

Moderate English editing is required.

7.Are the conclusions consistent with the evidence and arguments presented?

Yes, the conclusions consistent with the evidence and arguments presented but further studies are required to confirm Authors’ hypothesis.

8.Do they address the main question posed?

Yes, the Authors addressed the main question posed.

Other comments:

  • English language: Moderate spell check required
  • Summary of abbreviations required.
  • Introduction: The Authors may improve this section on the theme of alternative instruments for implant site preparation and placement. Allow me to suggest a relevant references to include: “https://doi.org/10.3390/jpm12010108”.
  • Materials and methods: This section has been properly prepared.
  • Results: This section has been properly prepared.
  • Discussion: What is the main theme that emerges from the authors' analysis?Is the study design a limitation? Please improve.
  • Conclusion: Further studies are required to confirm Authors’ hypothesis.
  • Figures and Tables: Please improve figures and tables quality if possible. Some captures are in Spanish language. Please correct.

After making the indicated changes, the article may be suitable for publication.

Thanks for the opportunity to review this manuscript.

Author Response

First of all thank you for your suggestions to improve our article, below I attach the changes we have made based on your recommendations

English language: Moderate spell check required: The translator has double-checked the English translation of the text.

Summary of abbreviations required: Thank you for pointing this out. For example we have changed CBCT for Cone-beam computed tomography systems

Introduction: The Authors may improve this section on the theme of alternative instruments for implant site preparation and placement. Allow me to suggest a relevant references to include: https://doi.org/10.3390/jpm12010108: Thanks. We have added the article that you recommended to us in the discussion of our article.

“The review by Bennardo F. et al 25 studied the use of magnetic mallets in oral surgery and implant procedures in terms of tissue healing, surgical outcome and complication rate in comparison with traditional instruments, which were used in the present study. The results obtained proved that the survival rate of the implants was 98.9% in the groups performed with magnetic mallet and 95.42% in the control groups, which were performed with traditional instruments.

Materials and methods: This section has been properly prepared.

Results: This section has been properly prepared.

Discussion: What is the main theme that emerges from the authors' analysis? Is the study design a limitation? Please improve.

Thanks for the note, we have added it, I attach the modification.

“A small sample size and a limited follow-up time were certain limitations of the present study.”

Conclusion: Further studies are required to confirm Authors’ hypothesis.

Thank you for your contribution, we have added this in conclusion.

“Despite the limitations of the present study, it may be concluded that.. “

Figures and Tables: Please improve figures and tables quality if possible. Some captures are in Spanish language. Please correct.  

Thanks for this note, we have made the necessary corrections with the language, here I enclose one of the tables so you can see it.

Reviewer 3 Report

Comments to the Author

Dear Authors,

I would like to thank you for your article titled “Analysis of Bone Loss in Supracrestal Implants with a Convergent Transmucosal Morphology”. The article needs to be revised in order to be accepted in International Journal of Environmental Research and Public Health. Please find the point-by-point comment from the reviewer.

  1. I would like to suggest revising the title. This study not only just focuses on the supracrestal implants, but also deals with three different kinds of depth of implant placement. Also, the title should mention that the present study is a retrospective clinical study.
  2. In the introduction, why the authors come up with the two hypotheses should be addressed: why the crestal implants were assumed to have a higher bone loss and why the infra- and supracrestal ones would not show significant difference compared to each other.
  3. The statement concerning STROBE guideline needed in the first paragraph of the M&M.
  4. In the M&M, the morphological characteristics should be addressed for each system in detail with the photograph if possible.
  5. In the Tables 1, 2 and 3, please name the first column of each table "number of implants".
  6. Like the authors provided the representative radiographs of the infracrestal group (Figures 1A, 1B and 1C), I would recommend provide additional representative figures showing the radiograph of crestal and supracrestal group at each time point. it will definitely make the readers understand better.
  7. Please address the limitation of the present study in the Discussion (i.e. small number of sample size).

Author Response

First of all thank you for your suggestions to improve our article, below I attach the changes we have made based on your recommendations

  1. I would like to suggest revising the title. This study not only just focuses on the supracrestal implants, but also deals with three different kinds of depth of implant placement. Also, the title should mention that the present study is a retrospective clinical study.

We have made changes in the title of our article, here below I show them to you:

“Analysis of perimplant bone loss with a Convergent Transmucosal Morphology: Retrospective clinical study”

  1. In the introduction, why the authors come up with the two hypotheses should be addressed: why the crestal implants were assumed to have a higher bone loss and why the infra-and supracrestal ones would not show significant difference compared to each other.

We put forward these hypotheses, because our theory was that thanks to the morphology of the transmucosal neck of the PRAMA and SHELTA+XA implants, characterized by having a cylindrical extension followed by a hyperbolic geometric curve, designed to guarantee true continuity with the abutment, it was the reason why they would show less bone loss compared to PREMIUM implants

  1. The statement concerning STROBE guideline needed in the first paragraph of the M&M.

Thank you for pointing this out. We have adapted the M&M, below I add some of the changes, although they are throughout all M&M:

2.1. Study design

A retrospective clinical study was carried out with a 1-year follow-up, with the aim of evaluating peri-implant bone loss after functional loading of implants, in both maxillary and mandibular premolar areas. The implants were placed by the same operator at the Dental Clinic of the Faculty of Medicine and Dentistry of the University of Valencia, between 2018 and 2019. The data were collected after signing the informed consent, approved by the Human Research Ethics Committee of the Universtity of Valencia (registration no.:1500285).

The sample used consisted of 30 implants placed in 30 patients. The sample was divided into 3 groups, infracrestal implants (n=10), crestal implants (n=10) and supracrestal implants (n=10).

The implants used were the following: (TABLES)

  1. In the M&M, the morphological characteristics should be addressed for each system in detail with the photograph if possible.

Thank you for your contribution, we have added this section in M&M

“ 2.2. Morphological characteristics of the implants”

  1. In the Tables 1, 2 and 3, please name the first column of each table "number of implants".

Thanks again, we have included it in the tables. One of the tables below is added, so you can see the modification.

SUPRACRESTAL (number of implants)

DIAMETER (mm)

LENGTH (mm)

1

3.8

11.5

2

4.25

8.5

2

4.25

10

4

4.25

11.5

1

5

10

  1. Like the authors provided the representative radiographs of the infracrestal group (Figures 1A, 1B and 1C), I would recommend provide additional representative figures showing the radiograph of crestal and supracrestal group at each time point. it will definitely make the readers understand better.

Thank you, we have added more images so that it can be clear to the readers.

  1. Please address the limitation of the present study in the Discussion (i.e. small number of sample size).

Thanks for the note, we have added it, I attach the modification.

“A small sample size and a limited follow-up time were certain limitations of the present study.“
